# 1-Hydroxy-2(1*H*)-pyridinone-Based Chelators with Potential Catechol *O*-Methyl Transferase Inhibition and Neurorescue Dual Action against Parkinson’s Disease

**DOI:** 10.3390/molecules27092816

**Published:** 2022-04-28

**Authors:** Joseph C. J. Bergin, Kean Kan Tan, Anya K. Nelson, Cristina-Andreea Amarandei, Véronique Hubscher-Bruder, Jérémy Brandel, Varvara Voinarovska, Annick Dejaegere, Roland H. Stote, David Tétard

**Affiliations:** 1Department of Applied Sciences, Faculty of Health and Life Sciences, Northumbria University, Newcastle upon Tyne NE1 8ST, UK; joecjbergin@gmail.com (J.C.J.B.); tankeankiat@hotmail.com (K.K.T.); anyanelson98@outlook.com (A.K.N.); 2Université de Strasbourg, CNRS, IPHC UMR 7178, F-67000 Strasbourg, France; amarandei@unistra.fr (C.-A.A.); veronique.hubscher@unistra.fr (V.H.-B.); jbrandel@unistra.fr (J.B.); 3Institut de Génétique et de Biologie Moléculaire et Cellulaire (IGBMC), Institut National de La Santé et de La Recherche Médicale (INSERM), U1258/Centre National de Recherche Scientifique (CNRS), UMR7104/Université de Strasbourg, 67404 Illkirch, France; varvara.voinarovska@helmholtz-muenchen.de (V.V.); annick@igbmc.fr (A.D.); rstote@igbmc.fr (R.H.S.)

**Keywords:** 1-hydroxy-2(1*H*)-pyridinone, catechol *O*-methyl transferase, Parkinson’s disease

## Abstract

Two analogues of tolcapone where the nitrocatechol group has been replaced by a 1-hydroxy-2(1*H*)-pyridinone have been designed and synthesised. These compounds are expected to have a dual mode of action both beneficial against Parkinson’s disease: they are designed to be inhibitors of catechol *O*-methyl transferase, which contribute to the reduction of dopamine in the brain, and to protect neurons against oxidative damage. To assess whether these compounds are worthy of biological assessment to demonstrate these effects, measurement of their p*K*a and stability constants for Fe(III), in silico modelling of their potential to inhibit COMT and blood–brain barrier scoring were performed. These results demonstrate that the compounds may indeed have the desired properties, indicating they are indeed promising candidates for further evaluation.

## 1. Introduction

Parkinson’s disease (PD) is a devastating disease characterised by the death of dopaminergic neurons in the substantia nigra pars compacta. In this very complex disease, ferroptosis, accumulation of iron and oxidative damage are thought to be critically involved in the pathology [1,2,3]. PD has no cure, but symptoms can be managed by the administration of L-DOPA as a precursor to dopamine. L-DOPA is capable of crossing the blood–brain barrier (BBB) into the central nervous system (CNS) where it is metabolised by aromatic aminoacid decarboxylase (AADC) enzymes into dopamine. However, L-DOPA and dopamine are further metabolised by catechol *O*-methyl transferase (COMT, E.C. 2.1.1.6), thus reducing the amount of dopamine available as a neurotransmitter [4]. To limit these unwanted metabolic degradations, inhibitors of AADC and COMT are usually co-administered with L-DOPA. In the case of COMT, inhibitors such as tolcapone, opicapone and entacapone (Figure 1) were developed and used in the clinic [4,5]. Tolcapone is rather unique in this set as it is the only COMT inhibitor that is known to significantly cross the BBB and act both peripherally and in the CNS [6,7]. These inhibitors are based on a nitrocatechol coordinating group and act by chelating the Mg^2+^ co-factor present in COMT [8,9]. One effect of the nitro group is to lower the p*K*a of the hydroxyl groups (around 4.6–4.7 and 10.0–10.2 in entacapone, nitecapone and tolcapone [10]), helping to bind to Mg^2+^. However, this nitro group has been linked to hepatic toxicity in tolcapone and its use has been discontinued [11].

We have recently described the use of metal chelators based on the 1-hydroxy-2(1*H*)-pyridinone (1,2-HOPO) coordinating group as neuron protective agents against toxins mimicking the insult observed in PD on dopaminergic neurons [16,17]. We proposed that their ability to bind excess Fe(III) and prevent oxidative damage to neurons contribute to their efficacy. Hydroxypyridinones are known to be good coordinating groups for hard Lewis acids such as Mg^2+^ and require only one deprotonation to be able to chelate a metal cation. These are expected to make 1,2-HOPO good alternatives to nitrocatechols in COMT inhibitors, without the associated hepatic toxicity.

We therefore hypothesised that simple derivatives of 1,2-HOPO could serve a dual purpose of COMT inhibition to limit dopamine degradation and neuron rescue against oxidative damage. Both of these are expected to be beneficial for treatment of PD. For neuron rescue, the compound must act in the CNS, requiring the compound to be able to cross the BBB. For this reason, we chose tolcapone as the template for modification (Figure 2). As the charge state of the compound is a key factor for BBB permeation, the p*K*a values of the compounds were determined and the BBB score, representing a statistical chance to cross the BBB, was calculated. Moreover, Fe(III) chelation ability is thought to be determining for the efficacy of the compounds as neuro-protective agents, as they could limit radical oxidation species (ROS) production via inhibition of the Fenton reaction. As such, their affinity for this metal and stoichiometry of the species formed were also determined. Finally, an in silico study was conducted to assess their potential as COMT inhibitors. The results reported herein show that they indeed have the potential to inhibit Fe(III)-catalysed oxidative damage, can inhibit COMT and are likely to penetrate the CNS. Further study of these compounds, including biological screening is therefore warranted.

## 2. Results and Discussion

The synthesis of compound **1a** is depicted in Figure 1. The synthetic strategy relied primarily on the formation of the 1,2-HOPO group in two steps from the 2-chloropyridine derivative, first forming the 2-chloropyridine *N-*oxide (**4a**, step ii). Subsequent substitution of the chlorine atom for a hydroxyl group (step iv) resulted in the formation of 2-hydroxypyridine *N-*oxide (**6a**) that is a tautomer of **1a**.

The formation of the ketone was planned in two steps. First was the formation of an alcohol using a Grignard reaction on commercially available aldehydes (step i) and a subsequent oxidation into the carbonyl group (step iii). We speculated that the oxidation needed to be performed after the formation of the *N-*oxide to avoid the possible formation of the ester via a Baeyer–Villiger oxidation when using a peroxyacid. It was deduced that the oxidation of the alcohol to the ketone was to be performed either after the 1,2-HOPO moiety had been fully formed or between the two steps required to synthesise that group. In the former option, this may have required the protection then deprotection of the 1,2-HOPO’s hydroxyl group to avoid possible interference with metal-based reagents used in the oxidation (e.g., MnO_2_) or with organic oxidation reagents (e.g., oxalyl chloride in the Swern oxidation). To limit the need for protection and deprotection, the oxidation of the alcohol was therefore planned between the formation of the 2-chloropyridine *N-*oxide (step ii) and its reaction with hydroxide (step iv).

The first step of the synthesis was therefore the reaction of a Grignard reagent from 4-bromotoluene on the aldehyde to form the alcohols **3a** in 93% yield. The formation of the 2-chloropyridine *N-*oxide **4a** using mCPBA resulted in the formation of a small amount of **5a**. We did not attempt to separate these compounds. Instead, the impure alcohol was oxidised into the diaryl ketone **5a** using manganese dioxide. Target compound **1a** was then obtained by reacting potassium hydroxide on the 2-chloropyridinine *N*-oxide.

The synthesis of compound **1b** followed a similar strategy with one exception (Figure 2). To our surprise, the formation of the pyridine *N-*oxide using mCPBA was also accompanied by the unexpected full oxidation of the alcohol into the wanted carbonyl group, thus giving **5b** directly and saving one step in our synthetic route. The unambiguous oxidation of the alcohol into the carbonyl was demonstrated by the appearance of a ^13^C NMR signal at 190.9 ppm, the expected MS fragmentations observed at *m*/*z* = 119.04927 (C_8_H_7_O, calc. 119.04969) and 91.05464 (C_7_H_7_, calc. 91.05478) and no evidence of any CH group in the ^1^H NMR around 4.5 ppm. We speculate that this rather mild and efficient oxidation might have been catalysed by traces of contaminant, possibly transition metals in the glassware (although this has been a systematic observation made with different glassware and different people) and proceeded via a radical mechanism [18].

The formation of compounds **5a** and **5b** in our synthetic strategy allowed us to use them as control compounds in our biological evaluations as they are expected to have similar electron distribution as **1a** and **1b**, but without the ability to bind Mg^2+^ or Fe^3+^. An additional control compound (**7b**, Figure 3), which is not expected to have any biological function [19], was prepared by the methylation of **1b** using methyl iodide in excellent purity without any need for specific purification procedures [20].

Physicochemical measurements were carried out on ligands **1a** and **1b** to evaluate their acido-basic behaviour and Fe(III) complexation properties in solution with respect to pH. The p*K*a of these compounds will have a strong influence on their ability to penetrate the brain [21,22]. It will also, together with the Fe(III) stability constant, influence their ability to redox-inactivate Fe(III), thought to be key for neuron rescue [23]. Acid-base properties of both ligands were determined by potentiometric titrations and UV-visible spectrophotometric titrations vs. pH (Figure 3). For solubility reasons, the studies were carried out in a mixed MeOH/H_2_O (80/20 *w*/*w*) solvent.

The titrations of both ligands showed the presence of one protonation equilibrium in the 2 to 12 pH range for each ligand. Analysis of the spectral variations [24,25] allowed us to calculate the protonation constants of the formed species (Table 1).

From these protonation constants, the electronic spectra (**1b** on Figure 3C and **1a** on Appendix A) and distribution curves (**1b** on Figure 4C and **1a** on Appendix A) of the species vs. pH were calculated. The distribution curves suggested that, at physiological pH (pH 7.4), both ligands are predominantly negatively charged (**1a** 76%, **1b** 100%) with the *N*-hydroxyl group of the 1-hydroxypyrydin-2(1*H*)-one ring being deprotonated.

As the ability to bind Fe(III) is a key factor for the evaluation of the potential use of these compounds against PD and oxidative damage, Fe(III) complexation studies vs. pH were carried out in the same way on compounds **1a** and **1b**, via spectrophotometric titrations between pH 2 and 12 (Figure 4).

The spectra of Fe(III) complexation titrations showed the appearance of a ligand to metal charge transfer (LMCT) band between 400 and 600 nm for both ligands, a sign that complexation started at very low pH. The LMCT band underwent a hyperchromic and hypsochromic shift with increasing pH suggesting an increase in the number of ligands in the coordination sphere of Fe(III), in accordance with the bidentate structure of the ligands. From a pH around 6, the LMCT band started decreasing and an increase in the baseline was observed, indicating the progressive decomposition of the complexes and precipitation of iron hydroxides (Fe(OH)_3_). Analysis of the spectral variations prior to precipitation suggested the successive formation of FeL_2_ and FeL_3_ complexes (L = **1a** or **1b**) in the pH range 2 to 12, in accordance with the bidentate character of the ligands. The calculated stability constants (log *β*) of the species are reported in Table 1. From the values of these stability constants, the electronic spectra of the complex species (Figure 4C and Appendix A) and their distribution curves (Figure 4D and Appendix A) were calculated. As the complexation of a metal by a ligand is dependent on several parameters such as metal ion concentration, ionic strength, ionic medium, pH, temperature and protonation constants of the ligands, the stability constants of the complexes cannot be used as such to compare the sequestering ability of a series of ligands for a given metal. The sequestering power of a ligand can be evaluated by the determination of the empirical and quantitative parameter pL_0.5_, which represents the total concentration of ligand required for the sequestration of 50% of the metal. [26]. A higher pL_0.5_ value will indicate a higher sequestering ability.

The pL_0.5_ values are presented in Table 1 (sequestering diagrams in Appendix A) and suggest that ligand **1b** is a substantially better Fe(III) chelator than **1a**. As a comparison, the 3,4-HOPO ligand DFP shows a pL_0.5_ value of 6.81. Previously studied 1-hydroxypyrazin-2(1*H*)-ones (1,2-HOPY) ligands **8** and **9** (structure in Appendix A), showed pL_0.5_ values of 7.34 and 5.98, respectively, and encouraging neuroprotection against the PD-relevant neurotoxin, 6-OHDA [27], which suggests that compounds **1a** and **1b** have a suitable Fe(III) chelation power and might help limit the formation of ROS in neurons.

Numerous methods have been described to predict the BBB permeation (via passive diffusion) of an organic compound [28,29,30]. Unfortunately, the molecular descriptors identified in these different methods as being key to BBB permeation are not always the same and some discrepancies exist, for example, regarding the influence of the partition coefficient log P usually taken as the n-octanol/water partition coefficient. Until a consensus is reached, we therefore have to assess the compounds using several tools to obtain a qualitative idea of their ability. The first tool we used was SwissADME [31]. Calculated parameters relevant to BBB Score of tolcapone, **1a** and **1b** obtained by SwissADME are given in Table 2.

Tolcapone is known to be able to cross the BBB [7] despite the negative prediction of SwissADME. Interestingly, our compounds are predicted to be BBB permeants by the same tool. In particular, **1a** and **1b** have far fewer numbers of hydrogen bond donors and acceptors. As the ability to cross the BBB has been proposed to be hampered by hydrogen bond formation [28], this suggests that our compounds are more likely to be BBB permeants than tolcapone, as required for neuron rescue.

Another recent method has defined the BBB Score based on the following parameters: number of aromatic rings (Aro_R), number of heavy atoms (HA), molar mass, number of hydrogen bond donors and acceptors, topological surface area (TPSA) and p*K*a [21]. The BBB scores of tolcapone, **1a** and **1b** are given in Table 3.

Tolcapone has a BBB score of 3.25 which gave it a 21.9% chance of being BBB permeant. Our compounds have a BBB Score of 4.8–5.0 that puts them in the higher category of compounds with a statistical chance of being BBB permeant of 54.5% and close to the category of compounds having a statistical chance of being CNS permeant of over 90%.

Another recent method to predict BBB permeation identified log P (partition coefficient of the neutral species, typically between n-octanol and water), distribution coefficient log D[pH 7.4] (typically the n-octanol/water distribution of the ionised species at pH 7.4), TPSA, neutrality/acidity/basicity, hydrogen bond donor (HBD), hydrogen bond acceptor (HBA), number of nitrogen atoms (NC), number of oxygen atoms (OC) and total number of nitrogen and oxygen atoms (NOC) as key parameters. [22]. The calculations for tolcapone, **1a** and **1b** are given in Table 4.

Again, this analysis indicates that our compounds are better than tolcapone with regard to HBA and HBD numbers. TPSA, OC and NOC are also much better although still classified in the red regions.

The p*K*as of tolcapone are 4.64 and 10.20. The p*K*a of our compounds are 6.9 and 5.69 for **1a** and **1b**, respectively. It has to be noted that the p*K*a values of compounds **1a** and **1b** were determined in a MeOH/H_2_O (80/20 *w*/*w*) solvent, which results in slightly higher values than in water. Thus, assuming that the p*K*a in water is not too much lower for our compounds compared to the ones measured in a mixed solvent, the percentage of non-ionised species present at physiological pH is far greater for compounds **1a** (24%) and **1b** (1.9%) than for tolcapone (0.2%), favouring brain penetration as required for neuron rescue, especially for **1a**. These properties suggest that our compounds could reach the CNS in a higher concentration than tolcapone.

Molecular modelling studies were carried out to investigate the nature of compounds **1a** and **1b** binding to COMT. More specifically, focus was on whether there are any major structural or energetic factors that would suggest non-binding of the ligands. Following the protocols described in the Methods (Appendix A), molecular dynamics (MD) simulations of the known inhibitor tolcapone in complex with the COMT enzyme for rat (crystal structure available) and human COMT (model constructed by comparative docking) were carried out. The structural stability and free energy binding scores were compared to those obtained by simulations of both rat and human COMT complexed to molecules **1a** and **1b** (models constructed by comparative docking).

Molecular dynamics (MD) simulations of tolcapone were carried out in two different protonation states and in a single orientation (based on the crystallographic data, see Figure 5), while simulations of compounds **1a** and **1b** were performed in a single protonation state but in two different orientations with respect to the SAM cofactor (see Figure 6). Representations of the structures of the different tolcapone and **1a** and **1b** complexes are given in Appendix A, respectively.

The root-mean-square-difference (RMSD) time series of the backbone atoms calculated from the trajectories were compared for the rat and human complexes (Figure 7). The results show that the global conformations do not deviate significantly from the initial reference structures, which demonstrates the general stability of the different complexes. Even though the RMSD time series for **1b** shows slightly higher values that the other compounds in rat COMT, the results suggest that compounds **1a** and **1b** do not introduce any strong interactions that would perturb the protein structure. To further confirm this, average structures were calculated over the final 10 ns of the simulations and the backbone RMSDs were calculated by-residue for both rat and human COMT, Appendix A, and the average structures were superposed, Appendix A, respectively. These results show that all helices maintain their secondary structure (Appendix A) and the main structural changes are limited to the loops in the vicinity of the ligand binding site (Appendix A). As suggested by the RMSD times series shown in Figure 7, the comparison of the average structures further supports the conclusion that compounds **1a** and **1b** do not introduce any significant structural changes. A closer examination of the ligand conformation in the binding pocket shows that in some of the simulations, the ligand moved from a bidentate coordination starting configuration to a more monodentate coordination of the Mg^2+^ ion. An earlier study also observed a movement from bidentate to monodentate coordination [33]. In this work, we do not attempt to resolve this question of mono- vs. bidentate coordination, rather we focus on comparing relative behaviours between the complexes of the different ligands.

The root-mean-square-fluctuations (RMSF) for the crystallographic complex of rat COMT with tolcapone were calculated from the MD simulations and compared those calculated from the experimental temperature factors (B factors) determined for the crystal structure. In Figure 8, a comparison of the backbone RMSFs averaged by residue shows good agreement in the overall pattern of flexibility. Only residues 170–190 (220–240 in human COMT) in the MD simulations show motions notably greater than those determined from crystallographic B factors. Using the symexp option in the Pymol program (ref: The PyMOL Molecular Graphics System, Version 2.5 Schrödinger, LL) to visualise the position of the enzyme in the crystal lattice, we conclude that the lower values of fluctuations observed in the experimental structure are due to crystal lattice contacts. Visualisation confirmed that, in the crystal structure, these residues form an alpha helix that is in contact with other proteins in the lattice. In the simulations, this region is solvent-exposed and displays a wider range of motions.

The RMSF pattern for the docked complexes with compounds **1a** and **1b** is overall similar to the complex with tolcapone.

From each simulation, we calculated an average free energy of binding score by the MM/GBSA method [34] using the CHARMM program combined with in-house programs. The MM/GBSA method is a rapid calculation of the free energy of binding based on the molecular mechanics force field and an implicit representation of the solvent environment. The calculation was repeated for 1000 structures extracted from each simulation. Here, we do not consider the absolute values of the numbers, but their values relative to that of the doubly deprotonated tolcapone (rat and human). The results are presented in Table 5. The absolute values from the MM/GBSA calculations are given in Appendix A.

The relative binding free energy scores with respect to doubly deprotonated tolcapone in rat and human COMT were calculated. Between singly and doubly deprotonated tolcapone, the doubly deprotonated form is the stronger binder because of favourable electrostatic interactions with the Mg^2+^ ion. In the simulations of rat COMT, molecules **1a** and **1b** showed binding free energies in the same range as singly or doubly deprotonated tolcapone, depending on the orientation. In the simulations of human COMT, the ligand showed more varied free energy scores, but one orientation of **1a** (**1a** N-O^−^→SAM) had a binding free energy score more favourable than the doubly deprotonated tolcapone.

Interestingly, one orientation of compound **1a** (**1a** = O→SAM) gave a positive free energy score in both rat and human COMT (Appendix A), which suggests that there may be some orientational preference in the binding of this compound to COMT. The large error bars reflect the strong electrostatic contribution to the binding free energy, which leads to large fluctuations in the binding free energy.

The results provide the first investigation into the potential binding capabilities of a new class of COMT inhibitors, compounds **1a** and **1b**. Besides the structural stability of complexes of COMT with these ligands, as compared to simulations of a complex for which the experimental structure is known, the energetic rankings of these ligands were in the same range as the known COMT inhibitor, tolcapone. The simulation results suggest the possibility of a preferred orientation in the ligand binding pocket for at least one of the compounds (**1a**). While further investigations will be necessary, these first results do not provide any evidence against the potential binding capabilities of these compounds to the COMT enzyme.

## 3. Experimental

All reagents were purchased from Sigma-Aldrich (St. Louis, MO, USA) and Fluorochem (Glossop, UK) and used without further purification unless stated. Solvents were obtained from Fisher Scientific (Waltham, MA 02451 USA) and were of either reagent or HPLC Grade and column chromatography was performed using Fisher Scientific silica 60Å (35–70 μm). THF was dried over 3Å molecular sieves for 48 h. Reactions were monitored with thin layer chromatography (TLC) performed on Merck plastic foil plates pre-coated with silica gel 60 F254. Infrared-red (IR) spectra were recorded using a SensIR Technologies Durascope diamond anvil cell mounted on a Perkin-Elmer Paragon 1000 FTIR Spectrometer (Texas, TX, USA), wavenumbers are reported in cm^−1^ and band intensities are described using the following abbreviations: w = weak, m = medium, s = strong. NMR spectra were recorded using a JEOL ECS400 Delta spectrometer (Tokyo, Japan) at frequencies of 399.78 MHz for ^1^H NMR and 100.53 MHz for ^13^C NMR. All chemical shifts are reported in parts per million (ppm) relative to a tetramethylsilane (TMS) internal standard. Chemical shifts are reported as follows: Shift (δ) in ppm (multiplicity, coupling constant (J) in Hz, normalised integral). Multiplets are described as follows: s = singlet, d = doublet, t = triplet and dd = doublet of doublets. High-resolution mass spectrometry was performed using a Q Exactive Mass Spectrometer (New York, NY, USA) under electrospray ionisation (ESI) operated in positive and negative ion modes as specified (scan range 85–800 *m*/*z*). All observed peak values are recorded as mass–charge ratio (*m*/*z*).

### 3.1. 2-Chloro-5-([4-methylphenyl]hydroxymethyl)-pyridine (***3a***)

Under an argon atmosphere, magnesium (0.99 g, 41 mmol) was added to a rapidly stirring solution of 4-bromotoluene (7.57 g, 44.3 mmol) in dry tetrahydrofuran (95 mL). The mixture was then stirred at reflux for three hours until all of the Mg had been reacted. The Grignard mixture was then cooled to 0 °C and a solution of 2-chloropyridine-5-carboxaldehyde (5.22 g, 36.9 mmol) in dry THF (40 mL) was then added dropwise. The reaction was stirred for a further thirty minutes at 0 °C and then quenched with water. The product was then extracted with ethyl acetate and the organic layer was then washed with brine, dried with magnesium sulphate filtered and evaporated under vacuum. The solid was triturated with petroleum ether and evaporated to yield 8.05 g of yellowish crystals (34.4 mmol, 93%). ^1^H NMR (400 MHz, CDCl_3_) δ 8.37 (d, *J* = 2.3 Hz, 1H), 7.65 (dd, *J* = 8.2, 2.75 Hz, 1H), 7.27 (d, *J* = 8.2 Hz, 1H), 7.22 (d, *J* = 7.8 Hz, 2H), 7.17 (d, *J* = 8.2 Hz, 2H), 5.82 (d, *J* = 2.8 Hz, 1H), 2.56 (d, *J* = 3.2 Hz, 1H), 2.34 (s, 3H). ^13^C NMR (100 MHz, CDCl_3_) δ 150.3 (quat.), 148.0 (CH), 139.9 (quat.), 138.5 (quat.), 138.2 (quat.), 137.2 (CH), 129.7 (CH), 126.6 (CH), 124.1 (CH), 73.4 (CH), 21.2 (CH_3_).



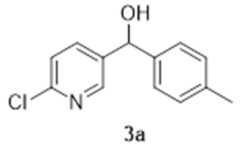



### 3.2. 2-Chloro-5-([4-methylphenyl]hydroxymethyl)-pyridine N-oxide (***4a***)

To a solution of **3a** (3.09 g, 12.4 mmol) in dichloromethane (50 mL), 10.32 g of *meta*-chloroperoxybenzoic acid (77 wt.%, 46 mmol) was added and stirred at reflux overnight. The organic phase was then cooled to 0 °C and then washed with a solution of sodium thiosulphate (10%, 3 × 50 mL) followed by a saturated solution of sodium bicarbonate and brine. The organic layer was then dried over magnesium sulphate and evaporated under reduced pressure to yield 0.75 g of a solid contaminated with **5a** which was used without further purification. ^1^H NMR (400 MHz, CDCl_3_) δ 8.35 (d, *J* = 1.3 Hz, 1H), 7.31 (d, *J* = 8.2 Hz, 1H), 7.12–7.21 (m, 5H), 5.70 (s, 1H), 4.70 (br, 1H), 2.32 (s, 3H). ^13^C NMR (100 MHz, CDCl_3_) δ 143.0 (quat.), 140.0 (quat.), 138.9 (CH), 138.9 × 2 (CH + quat.), 138.1 (CH), 129.5 (CH), 126.5 (CH), 126.3 (CH), 72.3 (CH), 21.2 (CH_3_). HRMS (Positive ESI), *m*/*z*: [MH]^+^ (C_13_H_12_ClNO_2_ + H^+^) calculated: 250.06348; observed: 250.06314.



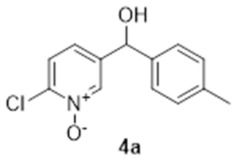



### 3.3. 2-Chloro-5-([4-methylphenyl]carbonyl)-pyridine N-oxide (***5a***)

To a solution of impure **4a** (ca 750 mg, ca 3.2 mmol) in dichloromethane (30 mL), manganese dioxide (5.2 g, 60 mmol) was added at 0 °C. The reaction mixture was stirred at room temperature overnight. The MnO_2_ was removed by filtration and the solvent was evaporated under reduced pressure to yield 670 mg of off-white crystals (2.71 mmol, 84%). ^1^H NMR (400 MHz, CDCl_3_) δ 8.68 (d, *J* = 8.2 Hz, 1H), 7.71 (d, *J* = 7.8 Hz, 2H), 7.64 (d, *J* = 8.2 Hz, 1H), 7.58 (dd, *J* = 8.7, 1.83 Hz, 1H), 7.33 (d, 2H, *J* = 8.2 Hz), 2.47 (s, 3H). ^13^C NMR (100 MHz, CDCl_3_) δ 190.8 (quat.), 145.3 (quat.), 145.0 (quat.), 141.7 (CH), 134.8 (quat.), 132.9 (quat.), 130.2 (CH), 129.7 (CH), 127.0 (CH), 126.1 (CH), 21.9 (CH_3_). IR: 1650 (C=O, s); 1292 (N-O, m). HRMS (Positive ESI), *m*/*z*: [MH]^+^ (C_13_H_10_ClNO_2_ + H^+^) calculated: 248.04783; observed: 248.04715.



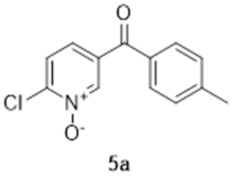



### 3.4. 1-Hydroxy-5-([4-methylphenyl]carbonyl)-2(1H)-pyridinone (***1a***)

A solution of aqueous potassium hydroxide (10% *w*/*v*, 7.5 mL) was added to a round bottom flask along with **5a** (0.670 g, 2.7 mmol) and stirred at 70 °C for 72 h. The reaction mixture was then cooled to 0 °C and concentrated hydrochloric acid was added dropwise until pH 1. The resulting precipitate was collected by Büchner filtration, washed with water and finally the crude product was recrystallised in ethanol to afford 232 mg of fine yellow needles (1.01 mmol, 37%). ^1^H NMR (400 MHz, CDCl_3_) δ 9.42 (s, 1H), 8.06 (d, *J* = 9.4 Hz, 1H), 7.38–7.43 (m, 4H), 6.80 (d, *J* = 9.4 Hz, 1H), 2.48 (s, 3H). ^13^C NMR (100 MHz, CDCl_3_) δ 187.1 (CH), 158.1 (quat.), 149.9 (quat.), 141.6 (quat.), 135.4 (CH), 130.1 (CH), 129.4 (CH), 122.8 (quat.), 116.0 (quat.), 114.8 (CH), 21.6 (CH_3_). HRMS (Positive ESI), *m*/*z*: [MH]^+^ (C_13_H_11_NO_3_ + H^+^) calculated: 230.08172; observed: 230.08101.



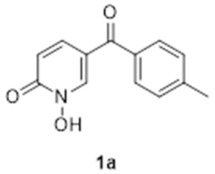



### 3.5. 2-Chloro-4-([4-methylphenyl]hydroxymethyl)-pyridine (***3b***)

To 95 mL of dry THF under an argon atmosphere, 0.99 g of magnesium turnings (41.2 mmol) and 7.61 g of 4-bromotoluene (44.5 mmol) were mixed and the mixture was heated to reflux for 2 h. The solution was then cooled to 0 °C and 40 mL of dry THF containing 5.0 g of 2-chloropyridine-4-carboxaldehyde (35.3 mmol) was added dropwise. The reaction was stirred at 0 °C for 30 min then quenched with 75 mL of water. The organic phase was separated and the aqueous phase was extracted 3 times with 50 mL of ethyl acetate. The combined organic phases were washed twice with 75 mL of water, dried over magnesium sulphate and evaporated. The solid residue was triturated with petroleum ether (60–80) then recrystallised in acetonitrile to give 4.69 g of the pure product (20.1 mmol, 57%). ^1^H NMR (400 MHz, CDCl_3_) δ 8.20 (d, *J* = 5.5 Hz, 1H), 7.40 (s, 1H), 7.13–7.21 (m, 5H), 5.68 (d, *J* = 3.7 Hz, 1H), 4.47 (d, *J* = 3.7 Hz, 1H). ^13^C NMR (100 MHz, CDCl_3_) δ 156.2 (quat.), 151.7 (quat.), 149.4 (CH), 139.2 (quat.), 138.5 (quat.), 129.7 (CH), 126.8 (CH), 121.6 (CH), 120.1 (CH), 74.4 (CH), 21.2 (CH_3_). HRMS (Positive ESI), *m*/*z*: [MH]^+^ (C_13_H_12_ClNO + H^+^) calculated: 234.06857 and 236.06562; observed: 234.06761 and 236.06429.



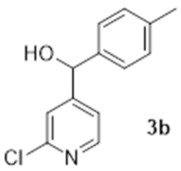



### 3.6. 2-Chloro-4-([4-methylphenyl]carbonyl)-pyridine N-oxide (***5b***)

In total, 3.0 g of **3b** (12.8 mmol) and 10.0 g of *meta*-chloroperoxybenzoic acid (77 wt.%, 44.6 mmol) were added to 50 mL of CH_2_Cl_2_ and the mixture was heated at 60 °C overnight. The solid that formed upon cooling of the solution was filtered and the organic phase was washed with three times 50 mL of saturated sodium hydrogencarbonate. The organic phase was then dried with magnesium sulphate and evaporated. The compound was purified on silica gel using 2.5% methanol in dichloromethane to yield 1.97 g of product (7.95 mmol, 62%). ^1^H NMR (400 MHz, CDCl_3_) δ 8.40 (d, *J* = 6.9 Hz, 1H), 7.93 (d, *J* = 2.3 Hz, 1H), 7.69 (d, *J* = 8.4 Hz, 2H), 7.63 (dd, *J* = 6.9, 2.3 Hz, 1H), 7.35 (d, *J* = 8.4 Hz, 2H), 2.47 (s, 3H). ^13^C NMR (100 MHz, CDCl_3_) δ 190.9 (quat.), 144.7 (quat.), 142.3 (quat.), 140.4 (CH), 133.8 (quat.), 133.0 (quat.), 129.9 (CH), 129.6 (CH), 127.7 (CH), 124.3 (CH), 21.8 (CH_3_). HRMS (Positive ESI), *m*/*z*: [MH]^+^ (C_13_H_10_ClNO_2_ + H^+^) calculated: 248.04783 and 250.04488; observed: 248.04700 and 250.04359.



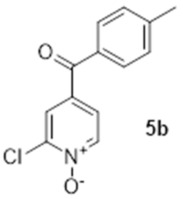



### 3.7. 1-Hydroxy-4-([4-methylphenyl]carbonyl)-2(1H)-pyridinone (***1b***)

A mixture of **5b** (0.32 g, 1.29 mmol) in 4 mL of a 10% potassium hydroxide solution was heated to 70 °C for 72 h. Upon cooling, a brown precipitate formed that was filtered. This solid was then redissolved in 5 mL of warm water and carefully acidified with concentrated HCl to pH 1. The yellow precipitate was filtered, rinsed with cold water to give 0.25 g of pure product (1.09 mmol, 84%). ^1^H NMR (400 MHz, CDCl_3_) δ 7.92 (d, *J* = 7.0 Hz, 1h), 7.74 (d, *J* = 8.0 Hz, 2H), 7.31 (d, *J* = 8.0 Hz, 2H), 6.98 (d, *J* = 2.0 Hz, 1H), 6.67 (dd, J = 7.0, 2.0 Hz, 1H), 2.45 (s, 3H). ^13^C NMR (100 MHz, CDCl_3_) δ 193.5 (quat.), 157.5 (quat.), 146.7 (quat.), 145.2 (quat.), 132.9 (quat.), 132.3 (CH), 130.5 (CH), 129.6 (CH), 118.8 (CH), 105.8 (CH), 22.0 (CH_3_). HRMS (Positive ESI), *m*/*z*: [MH]^+^ (C_13_H_11_NO_3_ + H^+^) calculated: 230.08172; observed: 230.08063.



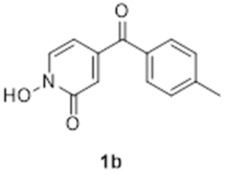



### 3.8. 1-Methoxy-5-([4-methylphenyl]carbonyl)-2(1H)-pyridinone (***7b***)

To 3 mL of dry DMF, 80 mg of **1b** (0.35 mmol), 170 mg of methyl iodide (1.20 mmol) and 150 mg of anhydrous potassium carbonate (1.08 mmol) were added and the mixture was stirred at room temperature overnight. The solid was then filtered, the solvent evaporated and water and ethyl acetate were added to the residue (10 mL each). The organic phase was separated. The aqueous phase was extracted twice with 10 mL of ethyl acetate and the combined organic phases were dried over magnesium sulphate and evaporated to give 54.4 mg of pure **7b** (0.22 mmol, 63%). ^1^H NMR (400 MHz, CDCl_3_) δ 7.75 (d, *J* = 8.0 Hz, 2H), 7.66 (d, *J* = 7.3 Hz, 1H), 7.30 (d, *J* = 8.0s Hz, 2H), 6.91 (d, *J* = 2.3 Hz, 1H), 6.48 (dd, *J* = 7.3, 2.3 Hz, 1H), 4.14 (s, 3H), 2.45 (s, 3H). ^13^C NMR (100 MHz, CDCl_3_) δ 193.4 (Cq), 157.8 (Cq), 147.2 (Cq), 145.0 (Cq), 135.8 (CH), 132.5 (Cq), 130.2 (CH), 129.4 (CH), 124.0 (CH), 104.5 (CH), 64.9 (CH_3_), 21.7 (CH_3_). HRMS (Positive ESI), *m*/*z*: [MH]^+^ (C_14_H_13_NO_3_ + H^+^) calculated: 244.09736; observed: 244.09732.



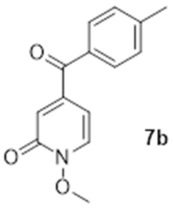



### 3.9. Physicochemical Studies

Water was purified via reverse osmosis, followed by passing through a mixed bed of ion exchange resin (Bioblock Scientific R3-83002, M3-83006) and activated carbon (Bioblock Scientific ORC-83005). All of the stock solutions were prepared by weighing solid products using an AG 245 Mettler Toledo analytical balance (precision 0.01 mg). Fe^3+^ cation stock solution was prepared from its perchlorate salt and its concentration was determined spectrophotometrically [35]. All the experiments were carried out at least in duplicate.

The acid-base properties (log *K*) of ligands **1a** and **1b** and their affinity for Fe^3+^ were determined via spectrophotometric titrations versus pH between pH 2 and 12. The titrations were carried out in a mixed MeOH/H_2_O 80/20 *w*/*w* (*I* = 0.1 M NaClO_4_) solvent due to their very low solubility in water. Sodium hydroxide (NaOH) and perchloric acid (HclO_4_) were used to adjust pH during titrations. The ionic strength of all the solutions was fixed to 0.1 M with sodium perchlorate (NaClO_4_). The measurement of pH was achieved by the use of combined glass electrodes (Metrohm, 6.0234.100, Long Life) filled with 0.1 M NaCl in MeOH/H_2_O (80/20 *w*/*w*). The electrodes were calibrated daily as hydrogen concentration probes by titrating known amounts of hydrochloric acid with CO_3_^2−^ free sodium hydroxide solutions. The GLEE program [36] was used for the glass electrode calibration with a p*Kw* of 14.42 for studies in MeOH/H_2_O (80/20 *w*/*w*).

Typically, an aliquot of 40 mL of ligand solution was introduced into a thermostated jacketed titration vessel (25.0(2) °C), with an additional 0.3 equiv of Fe^3+^ in the case of complexation titrations. A known volume of perchloric acid solution was added to adjust the pH value to around 2, and the titrations were carried out between pH 2 and 12 by addition of known volumes of sodium hydroxide solution with a Metrohm 904 DMS Titrino automatic titrator (Methrom AG; Herisau, Switzerland) equipped with a 2mL Dosino 800 burette. After each addition, the pH was allowed to equilibrate and a UV-visible spectrum was recorded automatically with an Agilent Cary 60 UV-visible spectrophotometer.

The spectrophotometric data were fitted with the HypSpec software [24,25], to calculate the protonation constants of the ligands (log *K*), the stability constants (log *β*) of the formed complex species and the coordination model of the studied systems. The data for Fe^3+^ hydrated species and their solubility products were taken into account in the equilibrium model.

### 3.10. Modelling Methods

Structures used in this study were Protein Data Bank entries 3s68 [9] (rat COMT + Tolcapone as inhibitor) and 5lsa [37] (human COMT + dinitrocatechol, DNC, as inhibitor). Rat and human S-COMT enzymes show a high degree of similarity; they differ by 40 amino acids, of which 3 are in the active site; these three are MET89(ILE139), MET91(ILE141) and TYR95(CYS145) in rat (human COMT numbering in parenthesis as given in the PDB files). All structures used for the simulations presented here were derived from these two structures. Protonation states of the proteins were determined using the PROPKA program [38].

In preparation of the structures for molecular dynamics simulations, crystallographic water molecules, including the one water molecule that coordinates the Mg^2+^ ion, were kept in the initial structure. Initial preparations were carried out using the CHARMM program [39].

The CHARMM36 all-atom force field was used for the protein [40] and the CHARMM generalized force field [41] was used to represent the small molecule ligands. The parameters for the ligands were obtained from the PARAMCHEM website [42] which generates parameters for novel ligands based on the CHARMM generalized force field. The partial atomic charges were then refined through quantum mechanical calculations using the Gaussian program [43]. To be consistent with the level used in the development of the CHARMM36 all-atom force field, the quantum calculations were carried out at the MP2/6-31G(d) level.

The system was energy minimised for 700 steps using the steepest descent algorithm with a non-bonded cutoff distance of 12 Å and the dielectric constant equal to 4. This minimisation had as its purpose to reduce any strong van der Waals clashes that may be present in the crystal structure. Following the initial minimisation, the system was solvated in an 85 × 85 × 85 Å side box of explicit TIP3P water molecules. Sodium and chlorine ions were added to the system to neutralise the charge and introduce a physiological ionic strength to the solution.

The NAMD program version 2.13 was used for the simulations which were carried out using the NPT isothermal–isobaric ensemble using the standard options in the NAMD program [42]. Periodic boundary conditions were used for all simulations. Long-range electrostatic interactions were accounted for the particle mesh Ewald (PME) method. The water molecules were first relaxed around the fixed protein by 1000 steps of Conjugate Gradient (CG) energy minimisation using a constant dielectric coefficient with ε = 1. Under the constraint of the fixed protein, the water and ions were heated to 600 K over 23 ps followed by 250 steps of CG minimisation, and finally heating over 25 ps to reach the temperature of 300 K. Next, the positional constraints were removed and the entire system was subject to 2000 steps of CG, followed by heating to 300 K over 15 ps. An integration time step of 2 fs was used for all simulations. Each simulation was run for 200 ns.

Calculations were carried out for rat and human COMT in complex with tolcapone, a known inhibitor of COMT. Simulations for rat and human COMT in complex with compounds 1a and 1b were also carried out. No experimental structures exist for these complexes, so they were built by comparative docking, i.e., using tolcapone or DNC as structural guides for placing **1a** and **1b** in the COMT binding sites.

## 4. Conclusions

In conclusion, we have designed and synthesised analogues of tolcapone that could constitute the first generation of a multi-functional treatment against the symptoms and progress of the disease. Modelling of the binding of Mg^2+^ in COMT indicated that they might indeed be inhibitors of the enzyme. These compounds are expected to act as COMT inhibitors in the CNS without its known side effects caused by the nitro group. Calculation of the BBB Score indicated that they are likely to penetrate the CNS. Calculation of their p*K*a and stability constant for Fe(III) suggests that they are suitable Fe(III) chelators and thus might limit iron-mediated formation of ROS. The biological evaluation of these molecules as COMT inhibitors and as neuro-rescue agents will be reported in due course.

## Data Availability

Data is contained within the article or Appendix A.

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
