# Peer review of "1-Hydroxy-2(1H)-pyridinone-Based Chelators with Potential Catechol O-Methyl Transferase Inhibition and Neurorescue Dual Action against Parkinson’s Disease"

_molecules, 2022, doi:10.3390/molecules27092816_

Round 1

Reviewer 1 Report

Before publication, a few problems should be solved.

(1) The goal of this research is not clear. It should be stated. A lot of effort has been put into in silico experiments and a lot of results has been devoted to it. However, the title says “synthesis”.

(2) Table 2 lists up ADME parameters. There are many ADME parameters which we can mention. Why did the authors select ADME parameters listed in Table 2? Can we say HBA and HBD belong to ADME parameters? The authors mentioned ADME parameters were obtained from SwissADME. As far as I know, SwissADME provides a lot of parameters.

(3) It is common for RMSD to be gradually converging over time. However, HOPO_1b N_SAM in Figure 7 shows reversed results.

(4) To distinguish comp’d 1a and 1b, NMR data described in section 4.4 and 4.7 should be assigned. For examples, which 13C chemical shift is ketone carbon of pyridinone.

Author Response

(1) The goal of this research is not clear. It should be stated. A lot of effort has been put into in silico experiments and a lot of results has been devoted to it. However, the title says “synthesis”.

The “Synthesis” word had been removed from the title. Initially, the first draft was going to focus only on the synthesis but the scope of the manuscript was expanded when more data became available before the submission deadline and we made a mistake by not correcting the title. Please accept our apologies. We have also rewritten part of the introduction to make our aims clear.

(2) Table 2 lists up ADME parameters. There are many ADME parameters which we can mention. Why did the authors select ADME parameters listed in Table 2? Can we say HBA and HBD belong to ADME parameters? The authors mentioned ADME parameters were obtained from SwissADME. As far as I know, SwissADME provides a lot of parameters.

We only selected the structural parameters that are relevant to BBB Score calculation. To avoid confusion, we have removed the term “ADME” and replaced it with the “Structural parameters relevant to BBB Score”.

(3) It is common for RMSD to be gradually converging over time. However, HOPO_1b N_SAM in Figure 7 shows reversed results.

We thank the reviewer for pointing this out.  In fact, in Fig. 7b the time series for HOPO_1b N_SAM was incorrected plotted.  The corrected Fig. 7 has been provided.  In Fig. 7a, the time series for HOPO_1b N_SAM is correct and, while the average RMSD is slightly higher for the HOPO_1b compound, the time series does not show any significant drift. We added a comment to the paper pointing out the slightly higher RMSD.

(4) To distinguish comp’d 1a and 1b, NMR data described in section 4.4 and 4.7 should be assigned. For examples, which 13C chemical shift is ketone carbon of pyridinone.

All the spectroscopic data on these two compounds are as expected, in particular, the 1H NMR are clearly conclusive by themselves and the assignment of the signals to each proton is fairly trivial. There are no similar compounds in the literature to compare our molecules to for further confirmation but the MS and 13C are also convincing. Even journals with the highest impact factors do not require a full assignment of the 13C signals or even assignment to C=O, Cquat, CH, CH2, CH3.

Reviewer 2 Report

Authors of “Synthesis of 1-Hydroxy-2(1H)-pyridinone-Based Chelators with Potential Catechol O-Methyl Transferase Inhibition and Neurorescue Dual Action Against Parkinson’s Disease” have done an interesting investigation that is suitable to be publish in this journal. Nevertheless, authors need to make some implementations and answer some questions.

  1. For the calculation of ADME parameters, they used the neutral or the ionic form of the molecules? The determination of these parameters using the two forms of the molecule will help to complement the physicochemical analysis of these compounds.
  2. When they use these ADME parameters to predict the BBB permeability they only used Tolcapone as reference. Therefore, they need to increase the number of molecules to which compare, at least five permeable and five non permeable compounds to BBB.
  3. Figures 5 and 6 need to be improve. Also, the 3D representation of these complexes must be added. This will increase the quality and presentation of the work.
  4. Did they perform a cluster analysis of the complexes obtained from the molecular dynamic’s simulations? A comparison of these structures will increase the discussion about the effect on the binding to COMT caused by the structural changes in Tolcapone.

Author Response

  1. For the calculation of ADME parameters, they used the neutral or the ionic form of the molecules? The determination of these parameters using the two forms of the molecule will help to complement the physicochemical analysis of these compounds.

Thank you for this useful suggestion. Our aim was to assess the BBB permeability of our compound as a justification for in-depth biological screening. The parameters are the ones of the neutral form as it is accepted that passive diffusion through the BBB is more likely for neutral molecules. The complex process of BBB penetration per se is compounded but the added complication of the equilibrium between neutral and anionic form. We estimated as a good approximation of the complex process that only the neutral form would start making its way through the BBB. Hence we estimate that if the neutral form is expected to show promising BBB permeability, this is a suitably good sign the compound are worthy of further biological studies, as per our aim.

  1. When they use these ADME parameters to predict the BBB permeability they only used Tolcapone as reference. Therefore, they need to increase the number of molecules to which compare, at least five permeable and five non permeable compounds to BBB.

It would be interesting to do this comparison if BBB permeation of many tolcapone analogues were known. To the best of our knowledge, this is not the case. We do not judge this comparison to be useful for molecules that are structurally very dissimilar to tolcapone. The authors who have developed the BBB Score have indeed already done such a work (the BBB Score is a statistical analysis of many permeable and non-permeable compounds of a range of structure). Our aim was to compare to tolcapone as it is the analogue we based our molecules on. Tolcapone is known to poorly be BBB permeant. As our compounds have better prediction of BBB permeability, it fulfils our need to demonstrate our compounds are worthy of further studies.

  1. Figures 5 and 6 need to be improve. Also, the 3D representation of these complexes must be added. This will increase the quality and presentation of the work.

Improved Fig. 5 and 6 have been provided and 3D representations of these complexes have been added to supplementary material. A phrase was added to the text indicating these 3D representations in the supplementary information.

  1. Did they perform a cluster analysis of the complexes obtained from the molecular dynamic’s simulations? A comparison of these structures will increase the discussion about the effect on the binding to COMT caused by the structural changes in Tolcapone.

We thank the reviewer for raising this point.  No, we did not do a cluster analysis because there are no major structural changes to the protein between the different compounds and we do not believe a cluster analysis would be very informative at this level.  However, to improve the discussion about the effects on ligand binding, we did go back to our data and compare average structures taken from the end of the simulations to discuss structural differences, which are essentially limited to the loops in the vicinity of the binding pocket.  3D representations have been provided as well as RMSD plots by residue